# Locomotor–Respiratory Entrainment upon Phonated Compared to Spontaneous Breathing during Submaximal Exercise

**DOI:** 10.3390/ijerph20042838

**Published:** 2023-02-06

**Authors:** Maja Marija Potočnik, Ian Edwards, Nejka Potočnik

**Affiliations:** 1Departmenet of Anasthesiology and Intensive Therapy, University Medical Center, 1000 Ljubljana, Slovenia; 2Centre for Cardiovascular and Metabolic Neuroscience, Department of Neuroscience, Physiology & Pharmacology, University College London, London WC1E 6BT, UK; 3Institute of Physiology, Medical Faculty, University of Ljubljana, 1000 Ljubljana, Slovenia

**Keywords:** locomotor–respiratory coupling, phonation, increased expiratory pressure, ventilatory efficiency, peak expiratory flow

## Abstract

Recently, increased attention to breathing techniques during exercise has addressed the need for more in-depth study of the ergogenic effects of breathing manipulation. The physiological effects of phonation, as a potential breathing tool, have not yet been studied. Thus, the aim of this study was to investigate the respiratory, metabolic and hemodynamic responses of phonated exhalation and its impact on locomotor–respiratory entrainment in young healthy adults during moderate exercise. Twenty-six young, healthy participants were subjected to peak expiratory flow (PEF) measurements and a moderate steady cycling protocol based on three different breathing patterns (BrP): spontaneous breathing (BrP1), phonated breathing pronouncing “h” (BrP2) and phonated breathing pronouncing “ss” (BrP3). The heart rate, arterial blood pressure, oxygen consumption, CO_2_ production, respiratory rate (RR), tidal volume (VT), respiratory exchange ratio and ventilatory equivalents for both important respiratory gasses (eqO_2_ and eqCO_2_) were measured (Cosmed, Italy) simultaneously during a short period of moderate stationary cycling at a predefined cadence. To evaluate the psychological outcomes, the rate of perceived exertion (RPE) was recorded after each cycling protocol. The locomotor–respiratory frequency coupling was calculated at each BrP, and dominant coupling was determined. Phonation gradually decreased the PEF (388 ± 54 L/min at BrP2 and 234 ± 54 L/min at BrP3 compared to 455 ± 42 L/min upon spontaneous breathing) and affected the RR (18.8 ± 5.0 min^−1^ at BrP2 compared to 22.6 ± 5.5 min^−1^ at BrP1 and 21.3 ± 7.2 min^−1^ at BrP3), VT (2.33 ± 0.53 L at BrP2 compared to 1.86 ± 0.46 L at BrP1 and 2.00 ± 0.45 L at BrP3), dominant locomotor–respiratory coupling (1:4 at BrP2 compared to 1:3 at BrP1 and BrP2) and RPE (10.27 ± 2.00 at BrP1 compared to 11.95 ± 1.79 at BrP1 and 11.95 ± 1.01 at BrP3) but not any other respiratory, metabolic or hemodynamic measures of the healthy adults during moderate cycling. The ventilatory efficiency was shown to improve upon dominant locomotor–respiratory coupling, regardless of BrP (eqO_2_ = 21.8 ± 2.2 and eqCO_2_ = 24.0 ± 1.9), compared to the other entrainment coupling regimes (25.3 ± 1.9, 27.3 ± 1.7) and no entrainment (24.8 ± 1.5, 26.5 ± 1.3), respectively. No interaction between phonated breathing and entrainment was observed during moderate cycling. We showed, for the first time, that phonation can be used as a simple tool to manipulate expiratory flow. Furthermore, our results indicated that in young healthy adults, entrainment, rather than expiratory resistance, preferentially affected ergogenic enhancement upon moderate stationary cycling. It can only be speculated that phonation would be a good strategy to increase exercise tolerance among COPD patients or to boost the respiratory efficiency of healthy people at higher exercise loads.

## 1. Introduction

Recently, attention to breathing techniques during exercise has increased among recreational and elite athletes. Whether the modification of breathing patterns (BrP) is possible without compromising the “minimal effort” homeostasis of the respiratory system requires discussion and more in-depth study. Breathing patterns differ according to the breathing frequency, breathing depth, inhalation/exhalation time relationship and maneuvers applied during expiration [1,2].

It is well known that the natural synchronization of the respiratory and locomotor systems provides a basis for the appropriate matching of ventilation to exercise performance [3], particularly during rhythmic activities such as running, walking, rowing, cycling and cross-country skiing [4,5]. Such synchronization is called entrainment. In swimming, conscious breathing manipulation is required to synchronize breathing with a suitable face position, in order to inhale air, and rhythmic arm swings [6]. The exact mechanism responsible for locomotor–respiratory coupling (LRC) is not known, although the literature suggests that the mechanical consequences of movement (e.g., the impact of the foot on the ground) affect the inertial motion of internal structures, such as the viscera and fat surrounding respiratory muscles, resulting in vertical rhythmic shifts of the diaphragm and exciting the peripheral muscle or joint receptors [5,7,8]. In humans, a wide variety of LRCs are applied, such as one breath per two, two and a half or three periodic movements of both extremities (1:2, 1:2.5, 1:3) [9]. The studies reported higher LRC in activities that involved arm movements [7,9] compared to isolated leg movements. However, during cycling, LRC is still present [5].

LRC is influenced by a number of factors, such as the type of exercise, fitness level, exercise intensity and presence of visual or auditory stimuli requires to maintain rhythm, as well as the voluntary initiative [10,11]. There are some reports stating that breathing/movement coordination improves overall performance [8] based on the reduced respiratory muscle work, delayed respiratory muscle fatigue, improved breathing efficiency, body stabilization during movement, eased airflow to and from the lungs [7] and reduced oxygen consumption (VO_2_) [8]. An improved breathing performance during exercise, which was related to spontaneous LRC, was reported by Bonsignore [10], while forced entrainment instigated no positive changes [12]. The percentage of entrained breaths was found to be significantly higher at loads under the gas exchange threshold [10] compared to higher loads.

In addition to LRC, expiratory maneuvers are believed to be particularly important for breathing efficiency, since increasing the resistance of the airways upon expiration can help in maintaining a positive expiratory pressure (PEP) until the end of exhalation, thus keeping the alveoli and airways open for longer during the exhalation period [13]. It is commonly believed that PEP improves alveolar ventilation, reduces breathing work and may potentially affect oxygen and CO_2_ kinetics during exercise [14]. To manipulate exhalation when exercising, pursed lip breathing [15], specially designed mouth guards [16] and jaw advancement splints [17] are often employed. During exhalation, airway resistance can be easily controlled by phonation [18]. Interestingly, a hearing stimulus operating in the frequency of the rhythm of the limb lowered energy expenditure upon cycling [5], and it was recently reported that rhythmic body movements yield impulses that result in entanglement between body motions, respiration and voice activities [19]. Additionally, locomotor forces are reported to have an enhanced expiratory effect if they occur when the abdominal muscles are recruited for forced expiration, as in phonation [18]. Thus, this study focuses on the physiologic response to exercise, integrating periodic movement, breathing and phonation. To the best of our knowledge, there are no reports about the potential physiological role of phonation during exercise and its interplay with LRC in improving exercise economy.

By exhaling against increased resistance, PEP is achieved, and it is commonly believed that PEP improves alveolar ventilation, at least in patients with pulmonary diseases [14] and potentially during exercise.

Thus, we aimed to test the extent to which airway resistance can be manipulated by the phonation of particular voices and whether phonated BrPs provoke favorable ventilatory responses to submaximal exercise compared to spontaneous breathing. The goal of our study was to examine whether the interaction between periodic movement (cycling), breathing and phonation in humans is significant enough to be physiologically important and impacts on exercise performance. For this purpose, the heart rate, arterial blood pressure, breathing frequency, tidal volume, oxygen consumption, CO_2_ production and other derived respiratory parameters were measured during moderate cycling under a constant moderate load and phonated BrP, with respect to spontaneous entrainment.

## 2. Materials and Methods

The study was performed in the Exercise Laboratory of the Institute of Physiology, Faculty of Medicine, University of Ljubljana. Ethical approval of the study was obtained from the National Ethics Committee (No. 0102-326/2018/5).

### 2.1. Subjects

To determine the sample size (software package, G*Power 3.1.9.2, Düsseldorf, Germany), the following input parameters were selected for the F test for the purpose of ANOVA: repeated measures, the within-between interaction with an effect size *f* of 0.314 (calculated from η^2^ = 0.1, determined by our preliminary measurements), a significance α level of 0.05, a statistical power (1 − β) of 0.8, 3 groups, 3 measurements, 0.5 as the correlation between repeated measurements and 1 for the non-sphericity correction. Therefore, we determined that 24 subjects would be sufficient (actual power = 0.828, critical F = 2.59) to assess the sought effects. To address the possible drop-out of some subjects, twenty-six subjects with comparable levels of physical activity were recruited upon public invitation to participate voluntarily in this crossover study. Their physical examination and history revealed no autonomic dysfunction, chronic diseases, medication usage or smoking. Their ECG and arterial blood pressure values were normal. Written informed consent was obtained before participation. The trial included 18 women and 8 men aged 20.85 ± 0.2 years old with a body mass index (BMI) of 22.97 ± 0.59 kg/m^2^.

### 2.2. Experimental Procedure

The study was carried out in a climate-controlled laboratory room between 9 and 12 am. The subjects refrained from physical exertion for at least 2 days before the first exercise test and were asked not to perform additional physical activities during the experiment period. They were not allowed to consume any alcohol, caffeine or tobacco for at least 2 h before the beginning of each exercise test and were asked to eat a light meal 1 h before attending the laboratory. Each participant visited the laboratory 4 times in February and March, with at least two days between the consecutive visits.

The level of expiratory resistance was manipulated by pronouncing different sounds which are known to obstruct the air flow upon expiration: the sound “ss” (as in the word “pressure”) and “s” (as in the word “subject”), with air flow obstruction in the mouth, and “h” (as in the word “how”), with air flow obstruction in the larynx, as compared to spontaneous breathing.

On the first visit, subjects were examined, their resting ECG was recorded, and their arterial blood pressure measured. The individual maximal heart rate (HRmax) was determined using the formula Hrmax = 205.8 − (0.685 age) [20], recommended as the most accurate formula for incremental exercise tests [21]. The subjects were familiarized with the experimental equipment and taught how to perform phonated breathing. After this, they were equipped with a silicon face mask for the gas flow and gas exchange analysis. The PEF was measured upon spontaneous exhalation as well as exhalation with phonating patterns. The PEF measurements were repeated three times in randomized order for each BrP. When recovered from the PEF measurements, the subjects performed a submaximal graded cycling test on a cycloergometer, starting at 30 W for three minutes as a warm-up period and continuing with a work-load increase of 30 W per minute until 85% of the participant’s HRmax was attained to determine the gas exchange threshold (GET).

Based on the PEF values upon pronouncing the sounds listed above, three BrPs were selected for use during exercise: spontaneous breathing (BrP1), “h” exhalation (BrP2) and “ss” exhalation (BrP3).

During the subsequent three visits, the subjects performed a 5 min aerobic cycling test at a constant load below the GET, applying different, randomly selected BrPs. Each session started with a blood pressure measurement at seated rest, and the breathing technique of the selected BrP was explained again. The oxygen consumption (VO_2_), CO_2_ production (VCO_2_), respiratory frequency (RR), tidal volume (VT) and cadence (C) were recorded in a breath-by-breath manner (Quark, Cosmed, Italy). The ECG in standard lead II and arterial blood pressure, employing the finger cuff for continuous blood pressure tracing, were assessed (Finapres 2300, Ohmeda, Madosom, WI, USA). The session consisted of 5 min of sitting at rest on the cycloergometer Ergoselect 100 (Ergoline, Germany) (baseline), 5 min of cycling at a predetermined load (cadence 60 ± 3 per minute) and 15 min of recovery after cycling cessation. During cycling, the selected BrP was applied, and at the end of the exercise, the rate of perceived exertion (RPE) was assessed using the Borg scale (Borg, 1982). The enhanced post-exercise oxygen consumption (EPOC) was determined using Quark CPET analyzing software (Cosmed, Italy) based on the work of Hughson and Morrissey [22].

### 2.3. Data Acquisition

The average of three accepted PEF results was calculated for each BrP, and the GET of each subject was determined by the V-slope method (Quark CPET Analysis). The breathing and metabolic variables were captured simultaneously on a breath-by-breath basis using Quark CPET hardware and software (Cosmed, Italy), while the arterial blood pressure and ECG were continuously recorded using the DATAQ system (DATAQ instruments Inc., DI-720 series, Akron, OH, USA) at 500 HZ. For the analysis, the last three minutes of cycling (exercise) were assessed. The oxygen consumption per body mass (VO2/kg), CO_2_ production per body mass (VCO2/kg), minute ventilation (VE), respiratory quotient (RQ), ventilatory equivalents for O_2_ (eqO2) and CO_2_ (eqCO2), oxygen pulse (O2pulse) and heart rate (HR) were assessed. The mean blood pressure was determined, and the heart rate recovery in 30 (HRR30) and 60 s (HRR60) was defined as the difference between the HR at the end of exercise and HR recorded 30 and 60 s after exercise cessation.

Based on the average RR during exercise, the subjects were enrolled in 9 intervals, ranging from 10 to 34 breaths per minute. The subjects were divided into the entrained (ent) and non-entrained (NONent) groups based on LRC, evaluated by examining the RR/C relationship during the last three minutes of exercise. The percentage of breaths occurring at rates corresponding to the integer ratio (N) ± 1 of the number of revolutions [10] was determined as 4 ≤ N ≤ 10, corresponding to RR/C = 1:2, 1:2.5, 1:3, 1:3.5, 1:4, 1:4.5 and 1:5, respectively. The subject’s breathing was considered entrained if the cumulative percentage of entrained breaths was higher than 60% [10]. The most frequent RR/C ratio was defined as characteristic for the given entrained subject. The entrained subjects were further distributed according to their characteristic RR/C results, and the most populated entrainment regime was defined as the most commonly used frequency ratio (MCUFR) (Jennifer M. Yentes 2019). Concerning all the BrPs, the subjects demonstrating MCUFR were enrolled in the MCURF entrainment group (entMCURF), and all other entrained (entO) subjects were placed in the entO group.

### 2.4. Statistical Analysis

The statistical analysis was completed using IBM SPSS Statistics, version 27 (IBM, New York, NY, USA). The data were tested for normality using the Shapiro–Wilk test and for the equality of error variances using Levene’s test. The possible violation of the equality of error variance assumption was considered as a limitation of the study. A level of confidence of *p* < 0.05 was selected. The paired t-test was used to compare the PEF results at different BrPs. The participants were divided into three groups in accordance with their entrainment regimes and pooled together, regardless of BrP. A two-way ANOVA, adjusted for gender as a covariate, was conducted, which examined the effects of entrainment and BrP on the respiratory, metabolic and cardiovascular variables, as well as the RPE (three entrainment groups, three BrPs). After the determination of the significant group and interaction effects, Tukey’s test was performed for the post hoc comparison. The data were analyzed for practical significance using magnitude-based inferences [23]. When a statistical difference was observed between groups, Cohen’s D (D) was calculated using the pooled standard deviation [24], and the thresholds for small, moderate and large standardized differences in the mean were set as D = 0.2, 0.5 and 0.8, respectively. The results are reported as the mean ± standard deviation.

## 3. Results

### 3.1. PEF Measurements

Upon phonation, the PEF was reduced compared to spontaneous breathing. The phonating BrPs demonstrated a gradually attenuated PEF in the order of “h”–“ss”–“s”, as shown in Figure 1. For further consideration, the “h” and “ss” BrPs were employed, since high airway obstruction upon “s” phonation can provoke lung hyperinflation by increasing the functional residual capacity and requires considerably increased breathing work, which may compromise safety during exercise performance [25].

### 3.2. RR and Entrainment during Exercise at Different BrPs

The distribution of the participants according to their RR and entrainment during exercise for all three BrPs is shown in Figure 2A–C. The breathing of 13 participants (50%) was entrained to cadence at BrP1 and BrP3, while 6 showed entrained breathing (46%) at MCUFR = 1:3 (Figure 2A,C). In BrP2, 11 (42%) participants were entrained, and six of them (55%) were entrained at MCURF = 1:4 (Figure 2B).

The average RR (18.8 ± 5.0) at BrP2 was lower compared to BrP1 (22.6 ± 5.5) and BrP3 (21.3 ± 7.2), though not significantly (F(2:112.6) = 2.714; *p* = 0.073; η^2^ = 0.073).

### 3.3. Other Respiratory and Metabolic Variables

There were no statistically significant interactions found between the effects of entrainment and BrP on any respiratory or metabolic variable studied (Table 1).

The main effect analysis showed that BrP affected the VT, which was significantly higher at BrP2 compared to BrP1 and 3, as seen in Table 1 and Table 2 and Figure 3. BrP seemed to affect EPOC, though not significantly. The EPOC was smaller at BrP2 compared to BrP1 and BrP3 (Table 1 and Table 2 and Figure 3).

Regardless of BrP, entrainment provoked significant differences in eqO2 and eqCO2, which were both significantly lower in the case of MCUFR entrainment (21.8 ± 2.2 and 24.0 ± 1.9) compared to the other entrainment coupling regimes (25.3 ± 1.9 and 27.3 ± 1.7) and when no entrainment was established (24.8 ± 1.5 and 26.5 ± 1.3), respectively. No interaction between phonated breathing and entrainment was observed (Table 2 and Figure 3) compared to the other entrainment regimes and non-entrained participants. Additionally, the participants in the MCUFR entrained group breathed at a significantly higher VT compared to the other two groups (Table 1 and Table 2 and Figure 3).

### 3.4. HR, MAP and RPE

There were no statistically significant interactions found between the effects of either entrainment or BrP on the HR, HRR30, HRR60, MAP or RPE.

Additionally, we found no main effects of BrP or entrainment on the HR, HRR30, HRR60 or MAP during moderate stationary cycling (Table 1 and Table 2 and Figure 3).

However, the participants found spontaneous breathing significantly less strenuous than either phonated breathing pattern (Table 1 and Table 2 and Figure 3).

**Table 2 ijerph-20-02838-t002:** The values of the measured respiratory and metabolic variables, heart rate, mean arterial blood pressure and RPE during moderate aerobic exercise based on different BrPs and entrainment regimes.

	BrP1			BrP2			BrP3		
	ent- MCURF	entO	NONent	entMCURF	entO	NONent	ent- MCURF	entO	NONent
HR (bpm)	137 ± 12	145 ± 21	144 ± 18	142 ± 11	136 ± 20	141 ± 24	132 ± 17	150 ± 19	136 ± 21
RR (min^−1^) *	20.2 ± 0.7	26.6 ± 1.0	21.4 ± 8.5	14.6 ± 0.7	20.9 ± 0.7	18.3 ± 6.1	20.1 ± 0.7	25.3 ± 1.1	19.6 ± 4.1
VT (L) *	2.0 ± 0.4	1.7 ± 0.2	1.9 ± 0.5	2.7 ± 0.3	2.2 ± 0.5	2.3 ± 0.5	2.0 ± 0.3	1.7 ± 0.2	2.1 ± 0.5
VE (L/min)	37.9 ± 5.8	41.1 ± 4.8	39.3 ± 6.9	37.3 ± 3.5	40.2 ± 6.9	38.0 ± 8.6	37.8 ± 8.0	38.0 ± 3.6	39.6 ± 7.5
VO2/kg (mL/min kg)	24.2 ± 3.1	23.1 ± 4.1	23.5 ± 2.5	23.4 ± 2.9	24.1 ± 3.1	23.5 ± 2.7	21.8 ± 3.6	24.4 ± 1.3	23.2 ± 2.4
VCO2/kg (mL/min kg)	21.2 ± 2.9	21.8 ± 3.9	22.3 ± 2.9	21.5 ± 3.4	22.2 ± 3.0	21.2 ± 2.4	19.7 ± 3.0	22.4 ± 1.6	22.4 ± 2.9
RQ	0.88 ± 0.04	0.93 ± 0.06	0.94 ± 0.08	0.92 ± 0.06	0.93 ± 0.03	0.95 ± 0.07	0.89 ± 0.06	0.92 ± 0.03	0.93 ± 0.08
eqO2 *	21.3 ± 1.1	25.9 ± 2.3	25.8 ± 4.6	21.9 ± 1.0	24.8 ± 3.2	24.3 ± 3.3	22.6 ± 2.4	24.9 ± 4.9	24.4 ± 3.0
eqCO2 *	23.7 ± 0.9	27.9 ± 1.5	27.5 ± 3.2	23.3 ± 1.0	26.7 ± 1.7	26.1 ± 3.9	25.0 ± 1.6	27.4 ± 3.5	26.2 ± 4.2
EPOC (mL)	764 ± 132	779 ± 70	718 ± 66	695 ± 126	724 ± 147	663 ± 154	793 ± 190	795 ± 207	720 ± 153
HRR30 (bpm)	25.1 ± 7.9	25.4 ± 6.7	23.6 ± 9.2	23.3 ± 9.3	27.1 ± 8.2	24.8 ± 8.4	32.4 ± 8.2	24.1 ± 8.7	23.8 ± 8.4
HRR60 (bpm)	37.3 ± 8.0	34.4 ± 11.0	38.7 ± 13.8	41.4 ± 11.2	41.0 ± 7.3	37.2 ± 9.5	44.8 ± 10.6	39.4 ± 5.5	45.4 ± 11.3
MAP (mmHg)	125 ± 28	128 ± 25	124 ± 18	112 ± 29	123 ± 11	128 ± 22	120 ± 23	133 ± 15	124 ± 30
O2pulse	12.0 ± 2.8	11.8 ± 3.9	10.2 ± 2.6	12.1 ± 1.6	12.3 ± 3.2	11.3 ± 3.5	12.3 ± 2.8	10.2 ± 3.3	12.4 ± 2.6
RPE *	10.13 ± 1.36	11.20 ± 0.45	9.88 ± 2.48	13.50 ± 1.51	10.83 ± 0.41	12.75 ± 2.25	11.63 ± 1.18	12.17 ± 1.33	12.13 ± 3.04

BrP—breathing pattern; BrP1—spontaneous breathing; BrP2—phonated, exhaling “h”; BrP3—phonated, exhaling “ss”; entMCURF—most commonly used frequency ratio; entO—other entrainments; NONent—not entrained; HR—heart rate; RR—respiratory rate; VT—tidal volume; VE—minute ventilation; VO2/kg—O_2_ consumption per body mass; VCO2/kg—CO_2_ production per body mass; RQ—respiratory quotient; eqO2—ventilatory equivalent for O_2_; eqCO2—ventilatory equivalent for CO_2_; EPOC—enhanced postexercise oxygen consumption; HRR30—HR recovery in 30s; HRR60—HR recovery in 60s; MAP—mean arterial pressure; O2pulse—oxygen pulse; RPE—rate of perceived exertion. * Simple main effects of BrP and/or entrainment.

## 4. Discussion

Our study aimed to determine the interaction between periodic movement, breathing and phonation and its importance for physiology and kinesiology in young, healthy humans. Our first main finding was that there is no significant interaction between the effects of locomotion–respiratory entrainment and phonated exhalation during short periods of moderate stationary exercise among healthy adults. Our second main finding was that phonation induced a medium to strong peak expiratory flow reduction in the spontaneous breathing of healthy volunteers. Our third main finding was that the spontaneous LRC changed in response to phonated breathing and significantly affected breathing efficiency during exercise. Our fourth finding was that the breathing pattern applied during exercise significantly influenced the RPE, tidal volume and respiratory rate but not minute ventilation or the metabolic and cardiovascular responses to exercise among young, healthy participants.

These conclusions are based on three observations: (1) no differences in the steady state ventilatory, metabolic and cardiovascular variables during moderate exercise regarding the interaction between BrP and entrainment were evident; (2) phonated exhalation pronouncing “h” and “ss” gradually reduced the PEF to 29 ± 6 % and 58 ± 10%, respectively, compared to spontaneous exhalation; (3) the most frequently used respiratory rate was significantly lower under moderate airway resistance compared to strong airway resistance and spontaneous exhalation among the entrained subjects during moderate aerobic exercise; (4) the ventilatory equivalents for O_2_ and CO_2_ were significantly reduced in the MCUFR group compared to the other entrained and non-entrained groups, regardless of BrP; and (5) no difference in minute ventilation was measured, but the tidal volume and breathing frequency differed regarding BrP and entrainment.

### 4.1. PEF and Phonation

To the best of our knowledge, this is the first report investigating the peak expiratory flow upon the phonation of different sounds. Schmidt and colleagues measured PEF during flow-controlled expiration [26]. Based on their PEF measurements, obtained using the adjustable flow regulator, it can be concluded that pronouncing “h” induced moderate airway obstruction, and “ss” induced strong airway obstruction. While the airway obstruction upon pronouncing “s” was even higher, we did not apply this BrP during exercise for safety reasons. Expiration against airway resistance resulted in increased expiratory pressure. Using the data of Schmidt et al., it can be deduced that “h” exhalation induced an end expiratory pressure of approximately 3, and “ss” exhalation induced 5.5 cm H2O, respectively. Increased expiratory pressure improves airway wall stability [14] and attenuates small airway collapse. Similarly, pursed-lip breathing [26] and wearing specific mouthguards [13] compromised expiratory flow, and both strategies are widely accepted for ergogenic enhancement. Thus, it seemed highly relevant to prove the possible positive effects of phonated breathing on exercise economics during exercise.

### 4.2. RR and Entrainment during Exercise at Different BrPs

The respiratory rate did not change significantly in response to the BrP applied. However, a clear shift toward a reduced RR at BrP2 was recognized. Interestingly, no tendency toward RR attenuation, compared to spontaneous breathing, was found at BrP3 during exercise, indicating that under high air flow obstruction, as in “ss” phonation, the effort of the expiratory muscles and RR may be increased by a neural reflex mechanism to prevent lung volume increase [16]. During exercise, the RR plays an important role as a strong marker of physical effort, which is even stronger than the other traditionally monitored physiological variables [27]. In the literature, there are conflicting reports about the impact of pursed-lip breathing and the wearing of mouthguards and jaw repositioning devices on the RR during exercise. Shulze and collaborators [28] reported no differences in the RR when using mouthguards in rugby. However, Garner [29] considered that a reduced RR during exercise serves as a marker of ergogenic efficiency when wearing jaw repositioning devices during exercise. Improved ventilatory efficiency, as a main effect of prolonged expiration during incremental exercise, was also reported by Matsomotu [30].

Many different harmonic couplings (2:1, 3:1, 4:1, 5:2, etc.) have been reported in humans [31]. However, a dominant coupling is defined as the most commonly used frequency rate. We found that dominant coupling occurred at LRC 1:3 and 1:4. To the best of our knowledge, there is no other literature about MCUFR during cycling at a predefined cadence. Studies concerning running and walking reported dominant coupling at RR/C = 1:2 [32,33], and this discrepancy could be related to different stride frequencies and voluntarily attained walking (running) speeds.

Dominant coupling shifted from 1:3 at BrP1 to 1:4 at BrP2 and back to 1:3 at BrP3. This result is partly in accordance with the findings of Tabary and Rassler [34], who reported that increased breathing resistance significantly prolonged the breath duration, with a self-evident shift towards a higher RR/C ratio at constant C. Again, this was only the case for “h” phonation and was not observed for “ss” phonation in our study, indicating the existence of optimal expiratory resistance, with positive effects on exercise performance.

### 4.3. Other Respiratory and Metabolic Variables

Our study failed to confirm the interaction between entrainment and PEP breathing. Phonated breathing (BrPs) affects breathing efficiency during exercise, as proposed by Pouw and Fuchs [19], who showed that both vocalization and periodic movements are positively related to respiratory kinematics. This discrepancy could be attributed to the fact that they studied the effect of upper limb movement, whereas in cycling, the legs move rhythmically and the upper limbs are fixed on the handlebars. Breslin [35] reported that spontaneous rhythmic breathing through a pursed mouth can affect the coordination of respiratory muscle recruitment, in a manner similar to entrained exercise, and improve ventilation while protecting the diaphragm from fatigue in COPD. Thus, we can conclude that the entrainment/PEP interaction is expressed in COPD but not in healthy subjects and that the intensity of the exercise was not high enough to provoke an insufficient oxygen supply, which is characteristic of COPB patients. Additional investigations should be conducted to prove this assumption.

Our study revealed that phonation provokes a considerable increase in airway resistance upon expiration and enables the conscious, controlled regulation of the expiratory flow, similar to pursed-lip breathing. We found a reduced RR and increased VT at BrP2 compared to other breathing modalities (BrP1 and BrP3) but no other significant changes in any other respiratory or metabolic measures upon phonated exhalation compared to spontaneous breathing during moderate stationary cycling. These findings are in line with previous studies that reported reduced breathing rates and an increased tidal volume during pursed-lip breathing among people with respiratory disorders [36,37]. However, in contrast to our findings, many other pulmonary benefits of pursed-lip breathing have been reported. Sakhei and colleagues [38] found that oxygenation and CO_2_ excretion are improved and respiratory work is reduced upon PEP provoked by pursed lips, while De Araujo [39] and colleagues found that this type of breathing reduces dynamic hyperinflation and improves exercise tolerance and O_2_ saturation during exercise in COPD.

Exercise, among healthy subjects, can be employed as a model of impaired pulmonary function, as increased energy demand during exercise places an additional load on the respiratory system due to the increased need for O_2_ and enhanced CO_2_ production, especially at higher exercise intensities. No beneficial effects of phonated breathing on respiratory efficiency and oxygen consumption observed in our study can be attributed to the fact that the applied intensity of moderate exercise was not high enough to reveal the advantages of PEP breathing. Thus, further studies should be conducted at higher loads in order to determine the potential benefits of phonated exhalation during exercise.

There are many studies about the effect of wearing mouthguards on exercise performance. Regardless of the type of mouthguard, the reported physiological benefit during exercise is approximately the same as obtained through pursed-lip breathing [13]. The evidence of physiological benefits resulting from the wearing of mouthguards during moderate exercise is equivocal. The results of some studies largely corroborate with ours, reporting no reduction in ventilatory or metabolic measurements upon wearing mouthguards during moderate exercise, while describing decreases in VO2/kg, VE and anaerobic energy turnover during heavy exercise [16,28]. On the other hand, other studies [13,40] reported enhanced breathing efficiency when wearing mouthguards in both aerobic and anaerobic performance.

Concerning oxygen kinetics, our results revealed that EPOC differed in response to the BrP applied during exercise, but not significantly, exhibiting a reduction upon “h” pronunciation compared to “ss” and spontaneous breathing. This result is partly supported by the findings of Stucky and coworkers [41], who reported that the synchronous recruitment of either intercostal or abdominal muscles, as in phonation [18,42], facilitates venous return and thus increases pulmonary perfusion and alveolar O_2_ transfer upon the onset of exercise. Increased oxygen transport above the metabolic requirement limited the oxygen deficit incurred [41]. According to this theory, “ss” exhalation will exhibit similar EPOC attenuation, which is, however, not evident. We may only speculate that a shorter breathing cycle duration at BrP3 will limit oxygen transport to the capillaries, regardless of the increased venous return.

Our subjects found exercise with spontaneous breathing statistically easier compared to the other two BrPs. These results are in accordance with the available literature [2,26], reporting that breathing discomfort increased with higher expiratory resistance. A decreased rate of perceived exertion at BrP1 can be related to the fact that the subjects breathe in their primary breathing pattern during spontaneous breathing [43]. One might speculate that ongoing training at different BrPs would attenuate the perceived discomfort. Bonsignore and colleagues [10] found that the asynchrony of exhalation with movement can cause a feeling of discomfort. However, we did not find any interaction between entrainment and BrP regarding the effect on RPE. These results may be explained by a study of Maclennan and colleagues [12], who found that individuals who were forced to use a predefined breathing pattern instead of spontaneous breathing did not experience that less effort was required while breathing.

One of the main findings of this study was the large decrease in eqO2 and eqCO2 with entrainment at MCUFR for all BrPs, indicating the increased ventilatory efficiency ni the case of dominant coupling. Our findings are in line with previous studies [2,10] that reported that spontaneous entrainment improves ventilatory efficiency during moderate-intensity cycling. Locomotor–respiratory coupling is often observed in humans during activities that involve impact loading with each foot strike, such as walking, running and cycling, and originates from mechanical and neurological interactions [5]. The breathing frequency can be continuously adjusted during locomotion through the combined activation of peripheral and central chemoreceptors and mechanoreceptors located in the joints and muscles in order to assist breathing under forces produced during locomotion [44,45] or by decreasing the energetic cost of lung ventilation [46].

Our study had some limitations. The participating subjects were not balanced regarding gender. As seen from Table 2, gender may have affected our results regarding the VT and VE but not eqO2 and eqCO2. The number of participants should be increased to exclude these effects. Furthermore, for VO2/kg and EPOC, a violation of the equality of error variance assumption was detected, yet these two parameters did not differ regarding either BrP or entrainment.

## 5. Conclusions

In this study, it was shown that the phonation of different sounds provoked graded alterations in expiratory flow obstruction related to positive expiration pressure breathing. Phonation during moderate stationary cycling affected the respiratory frequency, tidal volume and dominant locomotor–respiratory coupling but not any other respiratory or metabolic measures that would enable us to conclude that phonation augments ventilatory efficiency in healthy young adults. On the other hand, the ventilatory efficiency was shown to improve during moderate exercise for most of the commonly used cadence/breathing frequency ratios compared to the other LRCs and non-entrained cycling, regardless of phonation. No effects of the interaction between positive expiration pressure breathing and entrainment on any of the respiratory or metabolic variables studied were observed. This indicates that in young healthy adults, entrainment preferentially affects ergogenic enhancement during moderate cycling. It can be only speculated that phonation, as a simple tool used to increase expiratory resistance, would be a good strategy for increasing exercise tolerance in COPD patients or boosting the respiratory efficiency of healthy people at higher exercise loads and would demonstrate the positive interaction with entrainment. Further studies are needed to prove these assumptions.

## Figures and Tables

**Figure 1 ijerph-20-02838-f001:**
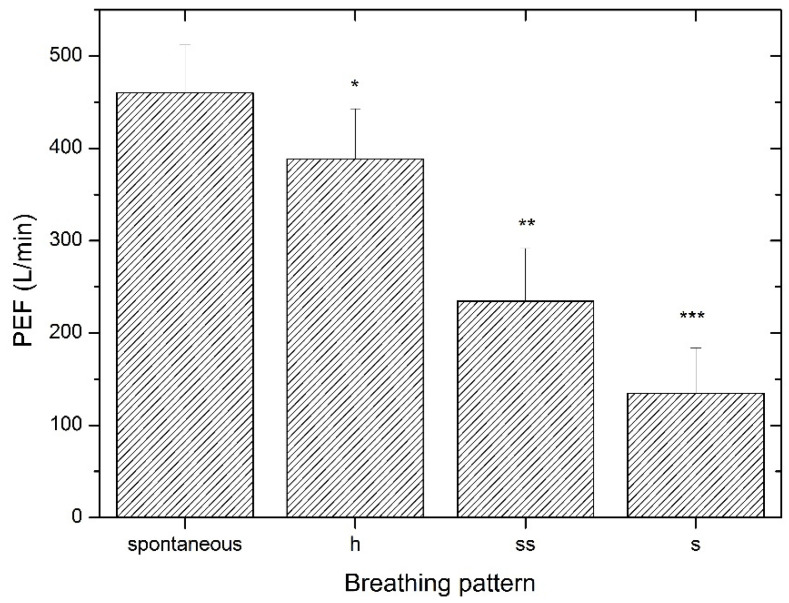
PEF at different breathing patterns. PEF—peak expiratory flow; spontaneous—spontaneous exhalation; h—phonated exhalation pronouncing “h”; s—phonated exhalation pronouncing “s”; ss—phonated exhalation pronouncing “ss”; *—significantly different compared to spontaneous breathing, ss and s; **—significantly different compared to h, s and spontaneous breathing; ***—significantly different compared to h, ss and spontaneous breathing. *p* < 0.05.

**Figure 2 ijerph-20-02838-f002:**
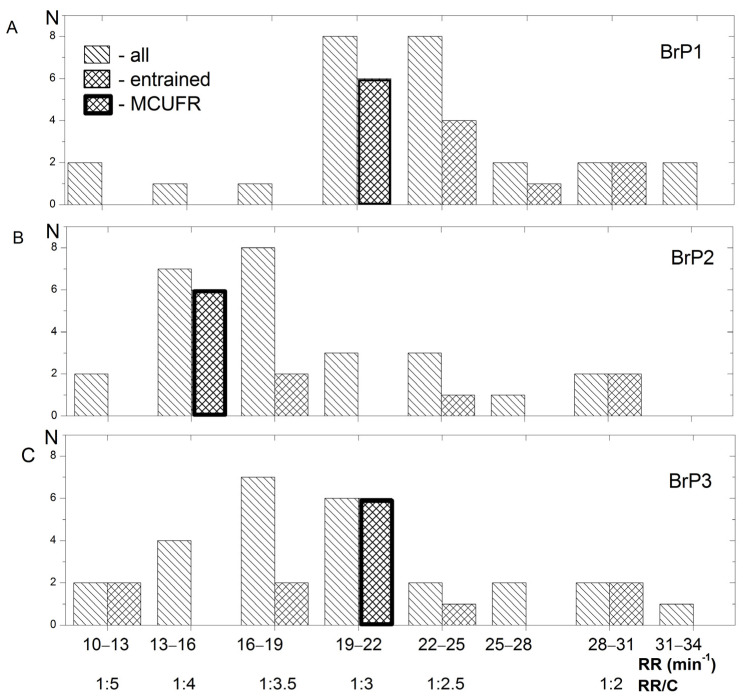
The distribution of the participants according to their RR and entrainment during stationary aerobic cycling. (**A**)—spontaneous breathing; (**B**)—breathing with phonated exhalation pronouncing “h”; (**C**)—breathing with phonated exhalation pronouncing “ss”; N—number of participants; RR—respiratory rate; RR/C—respiratory rate to cadence ratio; MCUFR—most commonly used frequency ratio; BrP1, 2, 3—breathing patterns.

**Figure 3 ijerph-20-02838-f003:**
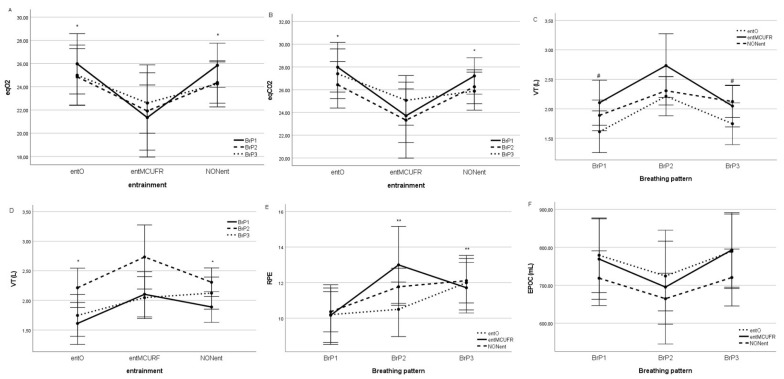
The changes in the measured respiratory and metabolic parameters, which differed significantly regarding either locomotor–respiratory entrainment or the breathing pattern. eqO2 for different entrainment groups (**A**), eqCO2 for different entrainment groups (**B**), VT for different breathing patterns (**C**), VT for different entrainment groups (**D**), RPE for different breathing patterns (**E**) and EPOC for different breathing patterns (**F**). VT—tidal volume; eqO2—ventilatory equivalent for O_2_; eqCO2—ventilatory equivalent for CO_2_; RPE—rate of perceived exertion (Borg scale); EPOC—enhanced postexercise oxygen consumption; BrP1,2,3—breathing patterns; entMCUFR—most commonly used frequency ratio; entO—other entrainments; NONent—not entrained. * Statistically significant differences in the entO and NONent groups versus entMCURF. ^#^ Statistically significant differences in BrP1 and BrP3 compared to BrP2. ** Statistically significant differences in BrP2 and BrP3 versus BrP1.

**Table 1 ijerph-20-02838-t001:** Statistical parameters of the interaction between the breathing patterns and entrainment with gender as a covariate and the main effects on the measured respiratory and metabolic variables, heart rate, mean arterial blood pressure and RPE during moderate aerobic exercise.

	BrP		Entrainment		BrP * Entrainment	Gender	
	*p*	η^2^	*p*	η^2^		*p*	η^2^
**HR**	0.701		0.960		0.626	<0.001	0.391
**VT**	0.001 *	0.175	0.023 ^#^	0.103	0.668	<0.001	0.176
	BrP1/BrP2: D = 2.90 large		entMCUFR/entO: D = 0.66 moderate				
	BrP2/BrP3: D = 0.65 moderate		entMCUFR/NONentr: D = 0.46 small				
**VE**	0.837		0.558		0.817	<0.001	0.378
**VO2/kg ^&^**	0.974		0.428		0.800	0.883	
**VCO2/kg**	0.782		0.292		0.922	0.502	
**RQ**	0.611		0.154		0.669	0.0419	
**eqO2**	0.814		0.014 ^#^	0.116	0.820	0.231	
			entMCUFR/entO: D = 1.17 large				
			entMCUFR/NONent: D = 1.00 large				
**eqCO2**	0.558		0.006 ^#^	0.139	0.688	0.400	
			entMCUFR/entO: D = 1.20 large				
			entMCUFR/NONent: D = 0.91 large				
**EPOC ^&^**	0.372		0.627		0.754	0.798	
**HRR30**	0.662		0.454		0.648	0.918	
**HRR60**	0.555		0.493		0.672	0.111	
**MAP**	0.587		0.694		0.966	0.432	
**O2pulse**	0.955		0.386		0.510	<0.001	0.758
**RPE**	0.011 *	0.145	0.525		0.549	0.183	
	BrP1/BrP2: D = 0.75 moderate						
	BrP1/BrP3: D = 0.91 large						

BrP—breathing pattern; BrP1—spontaneous breathing; BrP2—phonated exhaling “h”; BrP3—phonated exhaling “s”; entMCUFR—most commonly used frequency ratio; entO—other entrainments; NONent— not entrained; p—significance; η^2^—partial eta squared; D—Cohen’s D coefficient; HR—heart rate; VT—tidal volume; VE—minute ventilation; VO2/kg—O_2_ consumption per body mass; VCO2/kg—CO_2_ production per body mass; RQ—respiratory quotient; eqO2—ventilatory equivalent for O_2_; eqCO2—ventilatory equivalent for CO_2_; EPOC—enhanced postexercise oxygen consumption; HRR30—HR recovery in 30 s; HRR60—HR recovery in 60 s; MAP—mean arterial pressure; O2pulse—oxygen pulse; RPE—rate of perceived exertion (Borg).* Statistically significant regarding BrP. ^#^ Statistically significant regarding locomotor–respiratory entrainment; ^&^ violation of the equality of error variance assumption.

## Data Availability

The data that support the findings of this study are available from the corresponding author upon reasonable request.

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
