# Peer review of "Locomotor–Respiratory Entrainment upon Phonated Compared to Spontaneous Breathing during Submaximal Exercise"

_ijerph, 2023, doi:10.3390/ijerph20042838_

Round 1

Reviewer 1 Report

Overall an exciting study examines the respiratory, metabolic and hemodynamic responses of phonated exhalation and its impact on locomotor-respiratory entrainment in young, healthy adults at moderate exercise. The manuscript is well written. 

In the study participated, 18 females and 8 males. It is well known that sex and anthropometric characteristics influence oxygen consumption, CO2 production, respiratory rate, tidal volume (VT), respiratory exchange ratio and ventilatory equivalents; you should examine whether these characteristics affect the results. You can use them as covariates. 

Table 1 is confusing, and figure 3 is difficult to see the results. Generally, the figures are difficult to understand. Also, a sample size analysis would be necessary to conduct. 

The authors give perspectives on their results that COPD patients can use as an alternative rehabilitation method. 

Author Response

In the study participated, 18 females and 8 males. It is well known that sex and anthropometric characteristics influence oxygen consumption, CO2 production, respiratory rate, tidal volume (VT), respiratory exchange ratio and ventilatory equivalents; you should examine whether these characteristics affect the results. You can use them as covariates. 

Answer: We have conducted statistical analysis once again considering sex as a covariate and added related text in the statistical method section (see revised manuscript page 4/14, paragraph 2.4, row 196), the results regarding the influence of sex on measured parameters are added as new columns in the Table 2 and commented in the limitations of the study.

Table 1 is confusing, and figure 3 is difficult to see the results. Generally, the figures are difficult to understand. Also, a sample size analysis would be necessary to conduct. 

Table 1 will appear expanded over the whole page width in the final version so we believe it will not appear confusing any more, the same for Figure 3. For better understanding of the tables and figures we have implemented figure and table captions.

We have conducted sample size analysis as described in the revised manuscript on page 3/14, section 2.1, rows 109-116.

Reviewer 2 Report

This study describes how ventilatory efficiency improves with locomotion dominant in the respiratory coupling regardless of BrP compared to others entrainment or not entrainment coupling regimes. No interaction between phonic breathing and entrainment was observed during moderate cycling. It also demonstrates for the first time that phonation could be used as a simple tool for manipulation expiratory flow. Also significant results have shown that in healthy young adults, entrainment does expiratory resistance preferentially influenced the ergogenic potentiation over moderate stationary Cycling. On a clinical level, the study hypothesizes that phonation would be a good strategy to increase tolerance in COPD patients or to increase the respiratory efficiency of healthy people to greater resistance to exercise loads.

The manuscript can be published in its current form after correcting reference n° 21 with this citation: Robergs, R. A., & Landwehr, R. (2002). The surprising history of the" HRmax= 220-age" equation. Journal of Exercise Physiology Online5(2), 1-10.

Author Response

he manuscript can be published in its current form after correcting reference n° 21 with this citation: Robergs, R. A., & Landwehr, R. (2002). The surprising history of the" HRmax= 220-age" equation. Journal of Exercise Physiology Online5(2), 1-10.

We have corrected the reference 21 citation (see revised manuscript, page 14/14, row 516-517).

Reviewer 3 Report

Dear authors,

First of all, thank you for the opportunity to review this article.  The present studies whether phonated breathing affects respiratory, metabolic and haemodynamic efficiency during exercise.

In general, the article is clearly elaborated. It is a well conducted study; the aim and object of the research are understandable. The results are well presented, although the tables should be clearer. I would suggest you to improve them, as they are quite difficult to follow. One drawback of this study lies in the sample size, which should be clearly stated as a limitation by the authors.

In my humble opinion, this is a suitable work for the journal and especially for the special issue Physical and Sport Education, Physical Activity and Health Promotion. However, I would like you to better clarify some procedures:

When you break the sample in so many entrainment regimes, you end up with samples of N=4. Apparently, the entrainment regimes were all analysed together. It would be important to better clarify this aspect.

Regarding the statistical analysis, were the normality and homogeneity assumptions that would allow you to use the parametric analysis verified? I believe that it is important that you better clarify the statistical procedures that you used.

Finally, I would like to congratulate the authors for the work. I hope my comments and remarks helped!

Author Response

In general, the article is clearly elaborated. It is a well conducted study; the aim and object of the research are understandable. The results are well presented, although the tables should be clearer. I would suggest you to improve them, as they are quite difficult to follow.

Tables will appear expanded over the whole page width in the final version so we believe they will not appear confusing any more. For better understanding we have also implemented the table captions.

One drawback of this study lies in the sample size, which should be clearly stated as a limitation by the authors.

We agree with the reviewer, particularly because of the included participants were not balanced by gender. We have conducted sample size analysis as described in the revised manuscript on page 3/14, section 2.1, rows 109-116 and added this as a limitation as suggested (page 12/14, row 436).

In my humble opinion, this is a suitable work for the journal and especially for the special issue Physical and Sport Education, Physical Activity and Health Promotion. However, I would like you to better clarify some procedures:

When you break the sample in so many entrainment regimes, you end up with samples of N=4. Apparently, the entrainment regimes were all analyzed together. It would be important to better clarify this aspect.

Yes, all entrainment regimes were pooled together for statistical analysis. We have added this explicitly in the revised manuscript (page 4/14, paragraph 2.4, row 195-7).

Regarding the statistical analysis, were the normality and homogeneity assumptions that would allow you to use the parametric analysis verified? I believe that it is important that you better clarify the statistical procedures that you used.

Thank you for this comment. We have tested the data for normality by Shapiro-Wilk test. To check the equality of error variances assumption Levene’s test was employed. A possible violation of equality of error variances assumption were considered as a limitation of the study as suggested by SPSS. We have added these explanations in the revised manuscript (page 4/14, paragraph 2.4, row 192-4).